# The Use of an Inspiration-Synchronized Vibrating Mesh Nebulizer for Prolonged Inhalative Iloprost Administration in Mechanically Ventilated Patients—An In Vitro Model

**DOI:** 10.3390/pharmaceutics15082080

**Published:** 2023-08-03

**Authors:** Matthias Otto, Yannik Kropp, Evelyn Jäger, Michael Neumaier, Manfred Thiel, Michael Quintel, Charalambos Tsagogiorgas

**Affiliations:** 1Department of Anaesthesiology and Critical Care Medicine, University Medical Centre Mannheim, Medical Faculty Mannheim of the University of Heidelberg, Theodor-Kutzer-Ufer 1–3, 68165 Mannheim, Germany; 2Institute for Clinical Chemistry, Medical Faculty Mannheim of the University of Heidelberg, Theodor-Kutzer-Ufer 1–3, 68167 Mannheim, Germany; 3Department of Anaesthesiology, Emergency and Intensive Care Medicine, University of Göttingen, Robert-Koch-Str. 40, 37075 Göttingen, Germany; 4Department of Anaesthesiology, DONAUISAR Klinikum Deggendorf, Perlasberger Str. 41, 94469 Deggendorf, Germany; 5Department of Anaesthesiology and Critical Care Medicine, St. Elisabethen Hospital Frankfurt, Ginnheimer Straße 3, 60487 Frankfurt am Main, Germany

**Keywords:** nebulizers, aerosol drug therapy, iloprost, vibrating mesh nebulizer, aerosol deposition, mechanical ventilation, in vitro

## Abstract

Mechanically ventilated patients suffering from acute respiratory distress syndrome (ARDS) frequently receive aerosolized iloprost. Because of prostacyclin’s short half-life, prolonged inhalative administration might improve its clinical efficacy. But, this is technically challenging. A solution might be the use of inspiration-synchronized vibrating mesh nebulizers (VMN_syn_), which achieve high drug deposition rates while showing prolonged nebulization times. However, there are no data comparing prolonged to bolus iloprost nebulization using a continuous vibrating mesh nebulizer (VMN_cont_) and investigating the effects of different ventilation modes on inspiration-synchronized nebulization. Therefore, in an in vitro model of mechanically ventilated adults, a VMN_syn_ and a VMN_cont_ were compared in volume-controlled (VC-CMV) and pressure-controlled continuous mandatory ventilation (PC-CMV) regarding iloprost deposition rate and nebulization time. During VC-CMV, the deposition rate of the VMN_syn_ was comparable to the rate obtained with the VMN_cont_, but 10.9% lower during PC-CMV. The aerosol output of the VMN_syn_ during both ventilation modes was significantly lower compared to the VMN_cont_, leading to a 7.5 times longer nebulization time during VC-CMV and only to a 4.2 times longer nebulization time during PC-CMV. Inspiration-synchronized nebulization during VC-CMV mode therefore seems to be the most suitable for prolonged inhalative iloprost administration in mechanically ventilated patients.

## 1. Introduction

Acute respiratory distress syndrome (ARDS) is associated with a dysfunction of the pulmonary vasculature resulting in acute pulmonary hypertension with progressive right heart failure and the development of an acute cor pulmonale [1,2,3]. Various drugs are available for the treatment of acute right heart failure, many of which have been adopted from the treatment of primary pulmonary hypertension (PPH) and are administered orally or intravenously [4,5]. However, due to the lack of pulmonary selectivity, systemic administration is associated with side effects, such as a generalized vasodilation and pulmonary vasodilation in nonventilated lung segments, leading to a worsening of hypoxemia [6]. Pulmonary drug application can reduce these adverse effects, as has already been shown for the prostacyclin analogue iloprost.

In ARDS patients, an improvement in oxygenation was observed after nebulization of 5 µg iloprost, without a significant reduction in systemic blood pressure [7]. Nevertheless, there have been no studies that were able to demonstrate an improvement in patients’ outcomes [8,9]. One reason for this might be that in those trials, iloprost was administered via continuous nebulization [7,10]. The patients receive a large bolus of drug in a short period of time [11]. Since the half-life of iloprost ranges from several minutes to a maximum of 30 min, the therapeutic drug effect might be lost shortly after administration [12]. Therefore, prolonged intravenous iloprost infusion regimes have already been investigated in studies of patients suffering from severe PPH [13,14]. To avoid a systemic vasodilation in the often hemodynamically unstable ARDS patients, a prolonged inhalative administration regime would be more desirable. However, this has never been clinically evaluated, likely because in a clinical routine, prolonged inhaled aerosol administration is challenging due to a complex nebulizer setup [15]. A practical solution to this problem might be the use of inspiration-synchronized vibrating mesh nebulizers (VMN_syn_), which are already used to administer aerosolized antibiotics to mechanically ventilated patients [11,16,17]. By synchronizing aerosol delivery with the inspiratory air flow, a three- to nine-fold longer nebulization time can be achieved compared to bolus delivery, with similar or even higher drug deposition rates due to a minimized drug loss during expiration [11,17,18]. Unfortunately, there is no bench data comparing the nebulization time and the deposition rate of prolonged, inspiration-synchronized iloprost nebulization using a VMN_syn_ to continuous bolus nebulization using a VMN_cont_. It also remains unclear what effects different ventilation modes might have on drug deposition and nebulization time of a VMN_syn_, since the aerosol output of the pressure-triggered nebulizer might also be influenced by the pressure waveform of the ventilator, which can differ significantly depending on the ventilation mode [19].

Therefore, in an in vitro model of mechanically ventilated adults, the drug deposition rate and the nebulization time of an inspiration-synchronized vibrating mesh nebulizer (VMN_syn_:M-Neb flow+) and a continuous vibrating mesh nebulizer (VMN_cont_:Aerogen Solo) were compared in two basic ventilation modes: (A) volume-controlled continuous mandatory ventilation (VC-CMV) and (B) pressure-controlled continuous mandatory ventilation (PC-CMV).

## 2. Materials and Methods

### 2.1. Ventilator Circuit and Lung Model

A test lung (Michigan Instruments, Kentwood, MI, USA) was connected to an endotracheal tube (ETT) (Mallinckrodt COVIDien, Dublin, Ireland) with an internal diameter of 8.0 mm. According to the work of Ari et al., a low-resistance breathing circuit filter (Respirgard II, Vyaire Medical Inc., Mettawa, IL, USA) was placed in-line between the ETT and the test lung to collect the delivered iloprost [20]. To prevent contamination with expiratory air, the filter was bypassed during expiration using two one-way valves [21]. The ETT cuff was inserted into the housing of the bypass and then blocked tight to avoid leakage. The circuit was ventilated by a Datex Ohmeda Centiva/5 critical care ventilator (Salvia Lifetec GmbH & Co. KG, Kronberg im Taunus, Germany). The in vitro model and the expiratory bypass are shown in Figure 1.

### 2.2. Ventilator Settings

The ventilator was run both in VC-CMV and PC-CMV modes. The volume was set to 500 mL with a frequency of 15 per min. I:E ratio was set at 1:2, maximum inspiratory air flow was 40 L/min and PEEP was set at 5 cm H_2_O with peak inspiratory pressure ranging from 9 to 12 cm H_2_O, depending on the ventilation mode. Bias flow was set at 3 L/min.

### 2.3. Nebulizers

Since ventilator-integrated nebulizers are not generally available in every hospital, we tested two portable vibrating mesh nebulizers that can be used with any ventilator type. The Aerogen Solo with a USB controller (Aerogen Limited, Galway, Ireland) is a commercially available, continuous VMN. The M-Neb flow+ (NEBU-TEC International med. Produkte Eike Kern GmbH, Elsenfeld, Germany) is also a commercial VMN, which operates inspiration-synchronized. Comparable to the function of the Pulmonary Drug Delivery System (PDDS) previously described by Dhand et al., the control unit of the nebulizer is connected to the breathing circuit filter via a thin air pressure feedback tubing using the filter’s luer-lock connector [18]. Unlike the PPDS, however, aerosol delivery is not optimized to the first 75% of inspiration but throughout the entire inspiration phase as long as the nebulizer senses pressure changes. The tested nebulizers are shown in Figure 2.

### 2.4. Nebulizer Positions

The nebulizers were placed according to their already-known optimal position in the ventilation circuit: the VMN_syn_ between the ETT and the Y-piece adapter, and the VMN_cont_ 15 cm upstream on the inspiratory limb using extra tubing [18,20,22]. The nebulizer positions are also shown in Figure 1. The study protocol is shown in Figure 3.

### 2.5. Nebulization of Iloprost

To determine the volume in the nebulizer chambers, we used a gravimetric method presented by El Taoum et al., assuming that one milliliter of iloprost solution weighs one gram [23]. At the beginning of each test run, the empty nebulizer chamber was weighed (w_empty_) using a high-precision scale (Scaltec SBC 31, Scaltec Industruments GmbH, Göttingen, Germany) with 0.0001 g readability. To better measure subsequent changes in weight, the chamber was filled with 0.5 mL iloprost (Ventavis^®^, Bayer Schering Pharma AG, Berlin, Germany) with a concentration of 10 µg/mL (corresponding to 5 µg iloprost) and 2.5 mL of saline with a final iloprost concentration of 1.25 µg/mL and a total fill volume of 3 mL using a calibrated 1000 µL pipette (BRAND, Wertheim, Germany). The chamber was then reweighed (w_filled_). The nebulizer was then assembled and connected to the circuit using a T-piece (Aerogen Limited, Galway, Ireland). The nebulizers were tested 5 times in every position and every ventilation mode. The time was measured until the nebulizers no longer produced aerosol. After completion of nebulization, the chamber was weighed again (w_end_).

### 2.6. Iloprost Sample Preparation

To extract iloprost, the filter was carefully removed from the housing and added to a 30 kDa Amicon Ultra-15 centrifugal filter unit (Merck Milipore, Darmstadt, Germany). As an extraction solvent, 14 mL of isopropanol:methanol:ethanol (1:7:2, *v*/*v*/*v*, LC-MS Grade, Merck Darmstadt, Germany) was added to completely cover the filter and incubated in an ultrasonic bath (Sonorex RK100, 35 kHz, Bandelin, Berlin, Germany) for 10 min. The samples were centrifuged for 15 min at 4700 rpm (Multifuge X3R, Thermo Scientific, Schwerte, Germany). Internal testing showed that no more iloprost could be obtained following centrifugation. The chambers were washed with 7 mL of the extraction solvent. For the LC-MS/MS-Analysis, 195 µL of the eluate was transferred to a LC-vial and 5 µL of internal standard (50 ng/µL misoprostol, Sigma Aldrich, Darmstadt, Germany) was added.

### 2.7. Iloprost Analysis and Quantification

Iloprost filter deposition (m_filter_) was quantified by reversed-phase ultrahigh-pressure LC on an Ultimate 3000 HPLC (Thermo Scientific, Germany) with an analytical flow of 0.35 mL/min using a Hypersil Gold C8 column (3 μm, 2.1 × 100 mm, Thermo Scientific, Germany). The column was maintained at 60 °C. The mobile phase composition was 0.1% formic acid (FA, Sigma Aldrich, Darmstadt, Germany) in water (LC-MS Grade, Merck, Darmstadt, Germany) for solution A and 100% acetonitrile (LC-MS Grade, Merck, Germany) in 0.1% FA. The gradient was 1:25, 2:35, 2.5:45, 4.5:50, 6:80, 6.5:80 and 7:25 (time, %B). The LC system was coupled to an Iontrap amaZon speed (Bruker Daltonics, Bremen, Germany) via electrospray ionization source. The analytes were analyzed using multi reaction monitoring (MRM) with transmission of 378.3 > 325.3 for iloprost and 400.3 > 365.2 for the internal standard.

### 2.8. Iloprost Deposition Rate Calculation

The deposition rate of the nebulized iloprost was finally calculated as a quotient of the iloprost mass found in the filter (m_filter_) and the 5 µg of iloprost the nebulizers had been filled with:deposition rate of iloprost = m_filter_/5 µg(1)

### 2.9. Nebulizer Output Calculation

Assuming that one milliliter of iloprost solution weighs one gram, the nebulizer output was calculated as follows:nebulizer output = V_filled_ – Vend = w_filled_ − w_end_(2)

### 2.10. Statistical Analysis

Statistical analysis was performed using IBM SPSS Statistics 25. The iloprost deposition rate was stated as a percentage of the iloprost mass found in the collection filter and the 5 µg of iloprost loaded in the nebulizer chamber before the start of nebulization. Aerosol output per time was stated as mL per minute and nebulization time was in minutes and seconds. The data showed a Gaussian distribution. To compare differences in iloprost deposition, output and nebulization time between the two nebulizers, we used a two-sample *t*-test. Continuous variables were described using mean ± SD. Statistical significance was set at *p* < 0.05.

All graphs and figures were created using GraphPad Prism Version 6.

## 3. Results

### 3.1. Iloprost Deposition Rate

A direct comparison of the deposition rates of the two nebulizers during VC-CMV showed no significant differences (VMN_syn_: 44.4% ± 9.7% vs. VMN_cont_: 38.5% ± 4.2%; *p* = 0.293). During PC-CMV, however, the VMN_syn_ showed significantly lower deposition rates compared to the VMN_cont_ (33.5% ± 4.4% vs. 40.3% ± 4.0%; *p* = 0.033); see Table 1 and Figure 4.

During VC-CMV, the iloprost deposition rate of the VMN_syn_ was 10.9% higher than during PC-CMV (44.4% ± 9.7% vs. 33.5% ± 4.4%). This difference, however, was not significant (*p* = 0.05). The deposition rates of the VMN_cont_ also did not differ significantly between the two ventilation modes (VC-CMV 38.5% ± 4.2%; PC-CMV 40.3% ± 4.0%; *p* = 0.534).

### 3.2. Aerosol Output and Nebulization Time

Regardless of the ventilation mode, the VMN_syn_ showed a significantly lower aerosol output compared to the VMN_cont_ (VC-CMV: 0.04 mL/min ± 0.00 mL/min vs. 0.36 mL/min ± 0.04 mL/min; *p* < 0.001; PC-CMV: 0.09 mL/min ± 0.00 mL/min vs. 0.34 mL/min ± 0.03 mL/min; *p* < 0.001).

During VC-CMV, the aerosol output of the VMN_syn_ was more than twice as low as during PC-CMV (0.04 mL/min ± 0.00 mL/min vs. 0.09 mL/min ± 0.00 mL/min; *p* < 0.001). This led to a significantly longer nebulization time during VC-CMV, see Table 2 and Table 3 as well as Figure 5. The output of the VMN_cont_ did not differ significantly between the two ventilation modes (*p* = 0.753).

## 4. Discussion

Due to their short half-life, prostacyclines might benefit from a prolonged drug delivery [13]. Prolonged inhalative administration of nebulized drugs, however, poses a major challenge in clinical practice. Continuous drug infusion into the nebulizer chamber allows for aerosol treatment over a longer period of time [15]. This technique, however, requires a complex setup and is associated with high aerosol loss during expiration, inevitably leading to a significant increase in therapy costs, especially when expensive drugs such as prostacyclines are used [20,24,25,26,27]. In 2017, Rello et al. were the first to also recommend the administration of inhalative antibiotics by using an inspiration-synchronized vibrating mesh nebulizer (VMN_syn_) positioned between the endotracheal tube and the Y-piece [16]. However, this recommendation had the primary intention of maximizing the drug deposition rate, as inspiration-synchronized nebulization has been shown to significantly minimize drug loss during expiration [19]. In a bench model, the Pulmonary Drug Delivery System (PDDS), an investigational, non-commercial VMN_syn_, showed drug deposition rates of 50–70% [18]. Beneath the high deposition rates, another barely noticed key advantage of this nebulization technique is a three- to nine-fold longer nebulization time, resulting in a prolonged pulmonary drug administration, which would predispose inspiration-synchronized nebulization for prolonged drug administration. Therefore, we hypothesized that an inspiration-synchronized vibrating mesh nebulizer might be used for prolonged, drug-saving nebulization of iloprost in mechanically ventilated patients. We also hypothesized that in addition to the inspiratory time, there might be other ventilator-associated factors affecting inspiration-synchronized nebulization, such as the pressure waveform of the respirator, which has not been investigated so far.

The main findings of the study are: (1) Inspiration-synchronized nebulization during the VC-CMV mode seems to be most suitable for prolonged inhalative iloprost administration in mechanically ventilated patients. (2) Drug deposition and nebulization time are influenced by the ventilation mode. (3) The development of new, commercial VMN_syn_ should therefore focus on an optimized aerosol release during the initial part of the inspiration phase.

### 4.1. Iloprost Deposition Rate

The results of the present in vitro study provide evidence that the drug deposition rate of the VMN_syn_ is influenced by the ventilation mode. We found a considerable difference in the iloprost deposition rate between the two ventilation modes we evaluated in our bench model. During VC-CMV, the rate of the VMN_syn_ was almost 11% higher than during PC-CMV, however with a non-significant result (*p* = 0.50). These findings might be explained by the effects of the ventilation mode on the nebulizer, and in particular by the different flow patterns of the two ventilation modes. During PC-CMV, we observed that at the end of inspiration, the aerosol was no longer effectively moved forward through the endotracheal tube towards the collection filter simulating the lung. At the end of the inspiratory phase, larger quantities of aerosol were visible in the T-piece and in the endotracheal tube. This seemed to be due to the decelerating flow pattern of the PC-CMV mode. The drug-containing aerosol, which remained in the endotracheal tube and the T-piece, was then lost during expiration. This effect was not visible to the same extent when using the VC-CMV mode. The square waveform pattern of the VC-CMV mode achieves a constant high flow that seemed to transport the aerosol to the lungs with a constant efficiency until the end of the inspiration. This theory is supported by the fact that during our experiments when using VC-CMV, much less aerosol was visible in the T-piece at the end of the inspiratory phase.

In contrast to the results described above, the VMN_cont_ appeared to be completely unaffected by the ventilation mode, as iloprost deposition rates were nearly identical between the two ventilation modes. This is most likely due to the positioning of the nebulizer and the resulting reservoir function of the 15 cm tubing between the T-piece and the VMN_cont_, which has already been described by Ari et al. [20]. Even though the aerosol is not effectively transported towards the tube at the end of the inspiration due to the decelerating flow in PC-CMV mode, the tubing acts as an aerosol spacer chamber [20]. A large portion of this aerosol remains in the spacer so it is not lost during expiration, as with the VMN_syn_, and can be transported to the lungs during the next inspiration. Due to the placement of the VMN_syn_, however, this reservoir effect cannot be utilized here.

A previous article reported superior in vitro drug deposition rates of up to 70% when using an inspiration-synchronized vibrating mesh nebulizer system called PDDS [18]. Compared to the rates that Dhand et al. reported for the PDDS, the deposition rates of the VMN_syn_ we tested were considerably lower (VMN_syn_: 33.5–45.4% vs. PDDS: 50–70%) [18]. These results cannot be explained by the influence of the ventilation mode alone. A key difference between the PDDS and the VMN_syn_ we tested is that the PPDS delivers aerosol only during the first 75% of the inspiration phase, while the conventional VMN_syn_ (M-Neb flow+) produces aerosol throughout the entire inspiration [17]. By limiting aerosol release to a defined portion of the inspiration, the PDDS ensures that the dead space consisting of the upper airways, the endotracheal tube and T-piece does not contain any aerosol at the end of the inspiration that would then be wasted during expiration. In contrast, the VMN_syn_ lost all aerosol trapped in the dead space (endotracheal tube and expiratory bypass simulating the upper airways) during expiration, regardless of the ventilation mode. However, in the PC-CMV mode, this effect appeared to be exacerbated by the effects of the decelerating airflow described above, resulting in a significantly lower deposition rate when compared to the VMN_cont_.

Therefore, our bench model shows that the expected deposition rate when using a conventional VMN_syn_, in which aerosol release is not limited for a certain part of the inspiratory phase, tends to be higher during VC-CMV compared to inspiration-synchronized nebulization during PC-CMV. The deposition rate was affected not only by the waveforms of the ventilation modes, but also by the timing of aerosol production and release. As a consequence, the conventional VMN_syn_ was unable to outperform the iloprost deposition rate of the VMN_cont_. The development of new, commercial inspiration-triggered nebulizers should focus on optimizing aerosol delivery for only a defined period of the inspiration cycle to achieve consistent deposition rates regardless of nebulization mode and to emphasize the drug-saving potential of the nebulizer by further minimizing drug loss during expiration.

### 4.2. Aerosol Output and Nebulization Time

As already expected, the aerosol output of the VMN_syn_ was significantly lower compared to the output of the VMN_cont_. This led to a 7.5 times longer nebulization time during VC-CMV and a 4.2 times longer nebulization time during PC-CMV. Those findings are in line with the data published in the literature and demonstrate that a VMN_syn_ might very well be used for prolonged iloprost administration [17].

Remarkably, the aerosol output of the VMN_syn_ differed significantly between the two tested ventilation modes. During VC-CMV, the aerosol output was 2.3 times lower than during PC-CMV, leading to a significantly longer nebulization time. This could be explained by the influence of the different pressure waveforms of the ventilation modes on nebulizer triggering. During VC-CMV, a plateau phase occurs at the end of inspiration with both valves of the respirator closed. Due to the lack of pressure change during this phase, the VMN_syn_, which only releases aerosol when detecting a change in pressure, was not triggered. This led to the significantly lower aerosol output we observed. In the PC-CMV mode, aerosol is formed during the entire inspiration time leading to a higher aerosol output and a shorter nebulization time.

These results underline the major influence of the ventilation mode on prolonged drug nebulization when using a VMN_syn_. A solution to this problem might also be, as already described above, a limitation of the aerosol output to a defined part of the inspiratory phase. If aerosol generation for both ventilation modes were to end before the plateau occurs in the VC-CMV mode, the aerosol output and the nebulization time would be identical.

### 4.3. Advantages and Disadvantages of Parenteral and Inhaled Iloporost Administration in Mechanically Ventilated Patients

The main advantages of parenteral prolonged iloprost administration, an improved exercise tolerance and a higher rate of survival, have been demonstrated primarily for spontaneously breathing patients [13]. However, intravenous iloprost administration has major disadvantages regarding adverse effects, such as hypotension, gastrointestinal disorders, headache and a worsening of gas exchange [6,28]. Systemic hypotension and a worsening of gas exchange due to vasodilation in non-ventilated lung segments are highly undesirable especially in mechanically ventilated patients suffering from ARDS, as those patients regularly present with a combination of life-threatening hypoxemia and hypotension due to septic shock [6]. This might be the reason why there are no clinical studies available on intravenous iloprost treatment in ARDS patients. Only Dembinsky et al. compared intravenous vs. inhaled iloprost in an animal model of acute lung injury [29]. They found that only inhaled iloprost significantly improved pulmonary gas exchange [29].

The main advantage of inhalative iloprost administration in mechanically ventilated patients is a direct drug effect at the site of action with significantly fewer systemic side effects [7]. Because the aerosol only reaches ventilated parts of the ARDS lungs, intrapulmonary shunt formation with a worsening of hypoxemia is prevented. Additionally, in comparative studies, prostacyclines showed selective pulmonary arterial vasodilation almost identical to the vasodilatory effect of inhaled nitric oxide (iNO) [30,31]. The major disadvantage of pulmonary administration only seems to be the difficulty in establishing prolonged drug delivery, because up to now it has required a complex nebulizer setup using a continuous iloprost infusion into the nebulizer chamber [15]. All available clinical studies therefore seem to have used bolus inhalation [7,8,9,32].

### 4.4. Limitations

The results of the present work underline the need for in vitro models to study and to predict the deposition characteristics of inhaled drugs in mechanically ventilated patients when using different nebulization modes and ventilation modes. Byron et al. have already highlighted the importance of in vitro models for aerosol research and their strengths in comparative nebulizer technology studies [33]. However, the limitations of in vitro models must also be taken into account when interpreting the results.

Due to the in vitro design of our lung model, the delivered dose of iloprost might be overestimated regarding in vivo pulmonary drug deposition [33]. The Respirgard II filter cannot be regarded as an absolute filter. It is likely that despite our use of an expiratory bypass to prevent aerosol from contaminating the filter during expiration, similar to Fink et al., a small part of the aerosol might nevertheless have reached the collection filter, leading to slightly higher deposition rates [21,34].

There are factors found in vivo that would decrease aerosol deposition, such as the humidity of the lungs and airways, turbulent mixing in the upper airways, impaction on ramifications in the airways or atelectasis, which we did not simulate as it was not a focus of our bench study [34,35,36,37].

Also, the transferability of our in vitro results to critically ill, mechanically ventilated patients, who usually receive aerosolized iloprost, is limited. These patients are frequently treated using specific respirator settings to prevent hypoxemia, such as low tidal volume ventilation, high plateau pressure with high PEEP levels or balanced respiratory rates with alterations in duty cycles [38]. The I:E ratio is particularly important for inspiration-synchronized nebulization since aerosol is only generated during inspiration. The nebulization time is a function of the inspiratory time over the respiratory cycle’s total time [11]. A change in the ratio can therefore have a great influence on the nebulization time. Further bench studies are needed to determine the relation of inspiratory time and nebulization time.

## 5. Conclusions

In this in vitro model of a mechanically ventilated patient, a conventional inspiration-synchronized VMN (VMN_syn_) used in the VC-CMV mode achieved iloprost deposition rates comparable to those obtained with bolus administration using a VMN_cont_ while simultaneously showing a 7.5 times longer nebulization time, making this setup suitable for prolonged inhalative administration of iloprost in mechanically ventilated patients.

During the PC-CMV mode, however, the VMN_syn_’s deposition rate was considerably lower while the nebulization time decreased significantly, counteracting the intended prolonged and iloprost-saving drug administration. Therefore, the development of new, commercial VMN_syn_ should focus on an optimized aerosol release during the initial phase of inspiration. This way, the deposition rate and nebulization time become independent of the ventilation mode and the VMN_syn_ is able to unfold its drug-saving potential and achieve even higher deposition rates than the VMN_cont_.

Nevertheless, the effects of prolonged inhalative iloprost administration using a VMN_syn_ on pharmacokinetics and therapeutic drug effects still have to be investigated in clinical trials. A randomized cross-over study in mechanically ventilated ARDS patients might be able to reveal the effects of prolonged pulmonary iloprost administration on pharmacokinetics, pulmonary vascular resistance, right heart function, oxygenation and hemodynamics compared with those using bolus nebulization.

## Figures and Tables

**Figure 1 pharmaceutics-15-02080-f001:**
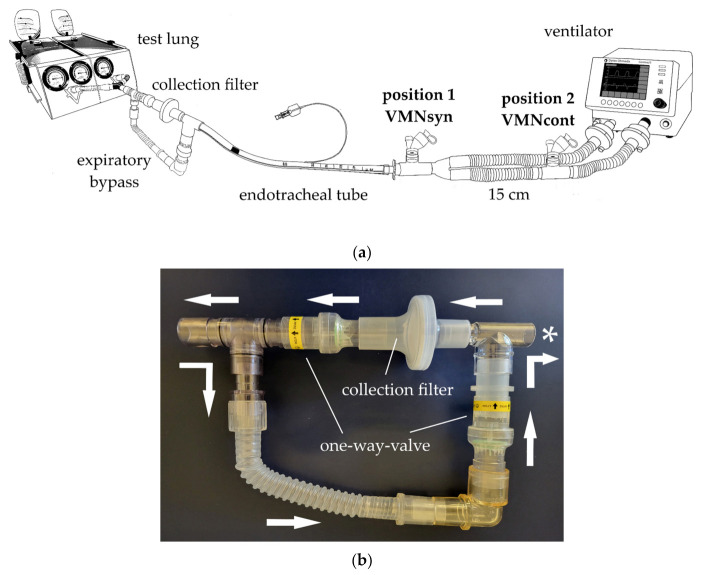
In vitro model of a mechanically ventilated patient: (**a**) the inspiration-synchronized VMN (VMN_syn_) was placed in position 1 between the endotracheal tube and the Y-piece, while the continuous VMN (VMN_cont_) was placed in position 2, 15 cm upstream on the inspiratory limb. (**b**) The filter was bypassed during expiration using two one-way valves. * Indicates the connection point of the endotracheal tube.

**Figure 2 pharmaceutics-15-02080-f002:**
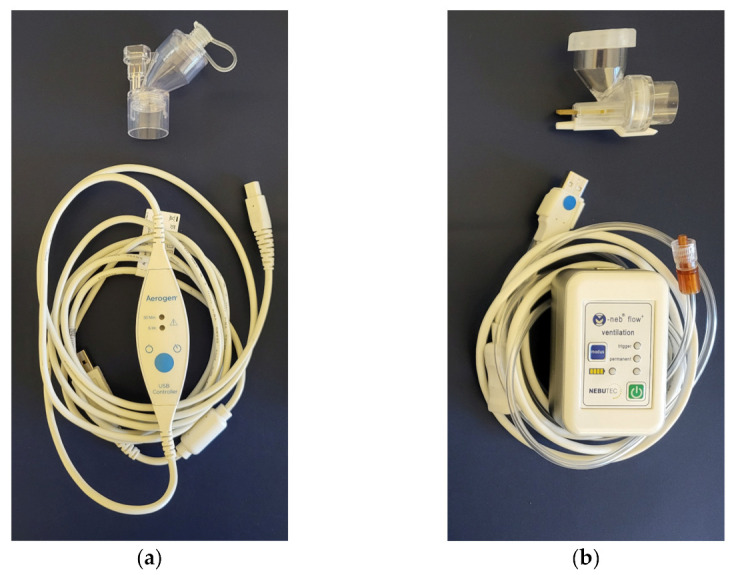
Aerosol generators. The two tested nebulizers were (**a**) a continuous mesh nebulizer (Aerogen Solo with USB controller) and (**b**) an inspiration-synchronized mesh nebulizer (M-Neb flow+).

**Figure 3 pharmaceutics-15-02080-f003:**
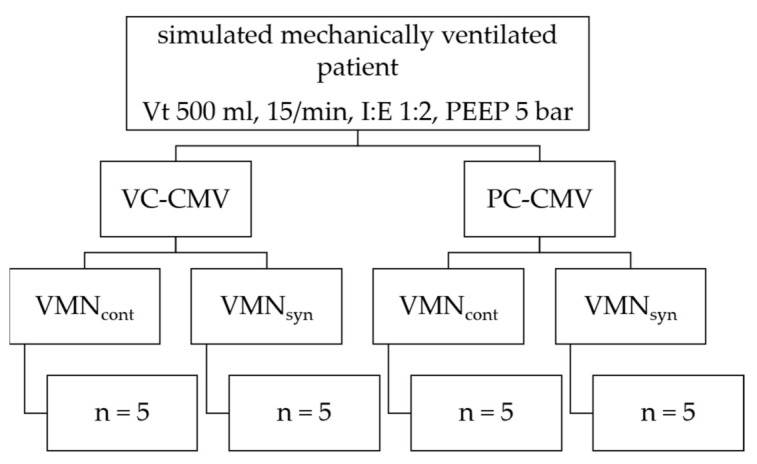
Study protocol. Hierarchy chart of the study protocol showing the different ventilation modes and nebulizers tested.

**Figure 4 pharmaceutics-15-02080-f004:**
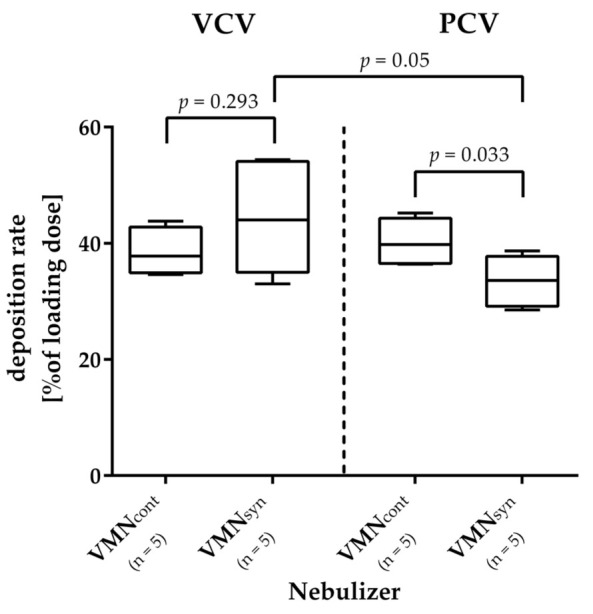
Iloprost deposition rate. The deposition rates of the two nebulizers during VC-CMV showed no significant differences, while during PC-CMV, the VMN_syn_ reached a significantly lower deposition rate. During VC-CMV, the iloprost deposition rate of the inspiration-synchronized VMN (VMN_syn_) was higher than during PC-CMV, however without being statistically significant (*p* = 0.05). The deposition rate of the continuous VMN (VMN_cont_) also did not differ between the two ventilation modes.

**Figure 5 pharmaceutics-15-02080-f005:**
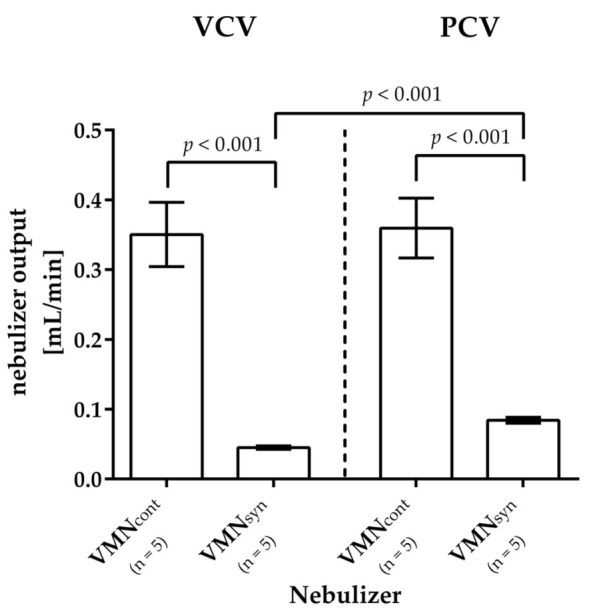
Aerosol output per minute. As expected, the VMN_syn_ showed a significantly lower aerosol output compared to the VMN_cont_ for both ventilation modes. During VC-CMV, the aerosol output of the VMN_syn_ was more than twice as low as during PC-CMV. The output of the VMN_cont_, however, did not differ significantly between the two ventilation modes.

**Table 1 pharmaceutics-15-02080-t001:** Iloprost filter deposition rates during volume-controlled ventilation (VC-CMV) and pressure-controlled ventilation (PC-CMV).

	Deposition Rate as Percentage of Nominal Dose (Mean ± SD %)		
Nebulizer	VC-CMV	PC-CMV	
VMN_cont_ (n = 5)	38.5 ± 4.2	40.3 ± 4.0	*p* = 0.534
VMN_syn_ (n = 5)	44.4 ± 9.7	33.5 ± 4.4	*p* = 0.50
	*p* = 0.293	*p* = 0.033	

**Table 2 pharmaceutics-15-02080-t002:** Aerosol output during VC-CMV and PC-CMV.

	Aerosol Output in mL per Minute (Mean ± SD %)		
Nebulizer	VC-CMV	PC-CMV	
VMN_cont_ (n = 5)	0.36 ± 0.04	0.36 ± 0.04	*p* = 0.753
VMN_syn_ (n = 5)	0.04 ± 0.00	0.09 ± 0.00	*p* < 0.001
	*p* < 0.001	*p* < 0.001	

**Table 3 pharmaceutics-15-02080-t003:** Total nebulization time during VC-CMV and PC-CMV.

	Nebulization Time in Min:Sec (Mean ± SD %)		
Nebulizer	VC-CMV	PC-CMV	
VMN_cont_ (n = 5)	8:26 ± 1:07	8:21 ± 1:05	*p* = 0.908
VMN_syn_ (n = 5)	65:40 ± 3:49	35:27 ± 1:38	*p* < 0.001
	*p* < 0.001	*p* < 0.001	

## Data Availability

All data presented in this study are included in the submitted manuscript.

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
