# Peer review of "The Use of an Inspiration-Synchronized Vibrating Mesh Nebulizer for Prolonged Inhalative Iloprost Administration in Mechanically Ventilated Patients—An In Vitro Model"

_pharmaceutics, 2023, doi:10.3390/pharmaceutics15082080_

Round 1

Reviewer 1 Report

The manuscript by C. Tsagogiorgas et al, describes the efficacy of two different methods of inhalative iloprost administration to address acute respiratory syndrome.

The study is carried on in vitro with a mechanical device used to quantify iloprost in a model of lung. The main drawback to publication is the exiguity of data gathered in the study. Many other authors have published on the inhalative use of Iloprost, mainly for the acute respiratory syndrome that is often associated with COVID and reported clinical parameters to evaluate efficacy.

In facts, the novelty in the manuscript relays the in vitro model to determine drug deposition, that could be useful for other applications.

Authors should address the need of an in vitro method to study inhalative drugs, and, more specifically, should give rational or speculation to the different results that they obtain in the different experimental setting (Table 1).

English should  be proof-read for spelling (for example, page 13, line 239).

Author Response

Dear Reviewer #1,

Thank you very much for your comments on our manuscript, which significantly improved our manuscript. Please find a detailed description of the revision in the attached letter.

On behalf of all co-authors

Your sincerely

Charalambos Tsagogiorgas

Reviewer 2 Report

In the current work, the authors have summarized their findings on administration of Iloprost via inhalation route and have used the vibration mesh nebulizer as the device for administration.  The authors have carried out exhaustive comparative study between different types of nebulizers in their distinct modes of operation and evaluated their performance in an in-vitro model of administration.  However, the manuscript needs minor edits with responses to the comments below:

1.       Please correct statements on lines 65 and 239 and replace the word “nice” to “nine” folds.

2.       Could you please include a discussion on the pros and cons of long-acting parenteral formulations of Iloprost post i.v. administration compared the inhalationally administered Iloprost for ventilated patients described in the current work.

3.       If we are moving the current developments to clinical trials, what amendments need to be made to the design? Please include this information with relevant references.

Author Response

Dear Reviewer #2,

Thank you very much for reviewing our manuscript. We appreciate all of your comments and suggestions, which helped us to improve the quality of the article.

Please find a detailed description of the revision in the attached letter.

On behalf of all co-authors

Your sincerely

Charalambos Tsagogiorgas

Round 2

Reviewer 1 Report

The manuscript has been significantly improved by the authors.